# Overcoming Vocabulary Constraints with Pixel-level Fallback

**Jonas F. Lotz**[*]
University of Copenhagen, Denmark &
ROCKWOOL Foundation Research Unit
jonasf.lotz@di.ku.dk

**Hendra Setiawan & Stephan Peitz**
Apple
{hendra, speitz}@apple.com

**Yova Kementchedjhieva**
MBZUAI, UAE
yova.kementchedjhieva@mbzuai.ac.ae

## Abstract

Subword tokenization requires balancing computational efficiency and vocabulary coverage, which often leads to suboptimal performance on languages and scripts not prioritized during training. We propose to augment pretrained language models with a vocabulary-free encoder that generates input embeddings from text rendered as pixels. Through experiments on English-centric language models, we demonstrate that our approach substantially improves machine translation performance and facilitates effective cross-lingual transfer, outperforming tokenizer-based methods. Furthermore, we find that pixel-based representations outperform byte-level approaches and standard vocabulary expansion. Our approach enhances the multilingual capabilities of monolingual language models without extensive retraining and reduces decoding latency via input compression.

## 1 Introduction

Subword tokenization is an intrinsic part of the modern language modeling pipeline (Schuster & Nakajima, 2012; Sennrich et al., 2016; Kudo, 2018). Tokenizers are trained to strike a balance between computational efficiency and vocabulary coverage. While larger tokenizer vocabularies offer better input coverage, the expanded embedding matrix significantly increases resource requirements. Consequently, language models typically adopt a moderate-sized vocabulary optimized for representational efficiency on the training corpus. Byte-level BPE (Wang et al., 2019; Radford et al., 2019) addresses the open vocabulary-problem, allowing, in principle, for the processing of any text without loss of information. However, fine-grained tokenization, down to the level of bytes, can lead to suboptimal performance, a problem particularly pronounced for languages and scripts that are underrepresented or absent from the training data (Muller et al., 2021; Rust et al., 2021; Pfeiffer et al., 2021).

The effectiveness of most large language models is constrained to English and a few high-resource languages (Touvron et al., 2023b; Jiang et al., 2023; Gemma Team et al., 2024), limiting the benefits of modern language technology for millions of users worldwide (van Esch et al., 2022). Meanwhile, English-centric language models possess latent linguistic capabilities applicable across languages (Brinkmann et al., 2025). A viable alternative to costly training on massive, multilingual data is thus to adapt pretrained English-centric models to new languages, leveraging their knowledge and capabilities (Peters et al., 2019).

Various approaches have been explored to extend language models to new languages and scripts, each with its drawbacks. *Vocabulary expansion* requires additional training to align new tokens with existing parameters (Wang et al., 2020; Chau et al., 2020; Lin et al., 2024), potentially at the cost of catastrophic forgetting (McCloskey & Cohen, 1989), especially after post-training steps such as supervised fine-tuning (SFT) or direct preference optimization

---

[*]Work done during an internship at Apple.

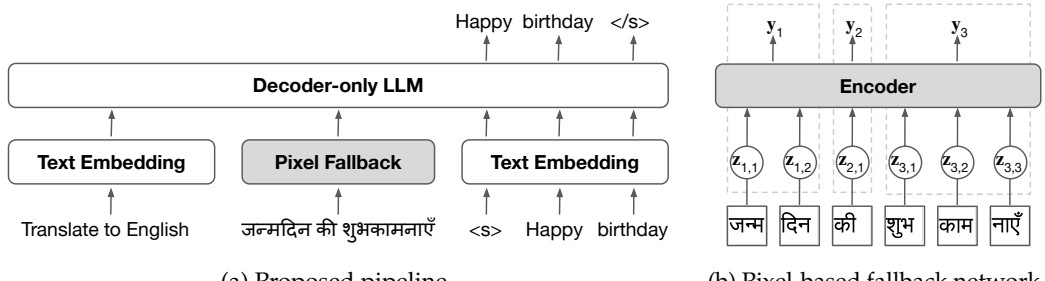

(a) Proposed pipeline.

(b) Pixel-based fallback network.

Figure 1: Illustration of our proposed NLP pipeline for Hindi-to-English machine translation. The decoder-only language model is instructed, encodes the source text using the fallback network, and autoregressively generates an English translation (left). Inside the fallback network the text is segmented into a list of words, rendered into image patches containing character bigrams, and projected into patch embeddings $\mathbf{z}_{i,j}$. The encoder outputs single-vector word representations $\mathbf{y}_i$, mapped as input embeddings to the language model (right).

(DPO). *Adapter modules* do not address the issue of suboptimal tokenization (Pfeiffer et al., 2020; 2021; Ansell et al., 2022). Finally, *transliteration* sacrifices the original representation and relies on heuristics which may not be available for all languages (Durrani et al., 2014; Muller et al., 2021; J et al., 2024). All of these methods operate within the vocabulary-based framework and as such remain limited by its constraints. We therefore propose augmenting the language modeling pipeline with a *fallback network*, which maps inputs suboptimally covered by the vocabulary directly into the embedding space of the language model (Pinter et al., 2017; Schick & Schütze, 2019), circumventing the tokenizer. We base our fallback network on the demonstrated effectiveness of pixel-based language encoding for vocabulary-free modeling where text is rendered to an image (Salesky et al., 2021; Rust et al., 2023; Lotz et al., 2023). Unlike recent approaches focusing on vocabulary embeddings (Gee et al., 2022; Dobler & de Melo, 2023; Liu et al., 2024b), the fallback network does not depend on complex heuristics or model-specific information. It is language-agnostic by design, and can be trained end-to-end jointly with any language model.

Since the fallback network exclusively improves input representations without modifying the vocabulary or output generation, we evaluate its effectiveness across tasks involving inputs in unseen scripts. We find that pixel-based fallback networks allow a 360M-parameter language model to exceed the performance of a 1.7B-parameter baseline and similarly push the 1.7B model beyond a 3.8B one. When trained on identical data, our pixel-based fallback network consistently outperforms standard vocabulary expansion and a byte-based fallback network. Additionally, the fallback network reduces inference time by up to $4\times$, particularly for larger language models and on languages prone to over-segmentation, by compressing input sequences. Strong transfer effects across visually similar scripts further emphasize the potential of pixel-based fallback networks for low-resource language modeling.

## 2 Proposed Approach

We propose to replace conventional input tokenization for unseen scripts with input embeddings generated by an external fallback network. Figure 1 exemplifies the proposed modeling pipeline in the context of machine translation with a decoder-only model. First, the language model is instructed with a prompt, which is embedded using the model's vocabulary. Next, the source text is rendered to an image and encoded by the fallback network. The concatenated representations from both the vocabulary and the fallback network are then passed to the decoder, which autoregressively predicts the English translation of the source text. Although our primary focus is on decoder-only architectures, we also evaluate fallback networks for encoder-only models, following the same logic of mapping inputs into the embedding space of the language model. Importantly, our approach treats the image-encoded source text the same as text embeddings, without converting it into discrete

tokens (Rolfe, 2017; van den Oord et al., 2017) or connecting the image encoder and the text decoder via layers of cross-attention (Alayrac et al., 2022; Li et al., 2023; 2024).

## 2.1 Fallback Network: A Vocabulary-free Encoder

Our fallback network is based on an encoder architecture that extends the Vision Transformer (ViT; Dosovitskiy et al., 2021) to text rendered as images, similar to PIXEL (Rust et al., 2023). Following ViT, the rendered image is split into patches $\mathbf{x} \in \mathbb{R}^{N \times (P^2 \cdot C)}$, where $N$ is the number of patches, $(P, P)$ is the resolution per patch, and $C$ is the number of channels. These image patches are then linearly projected into patch embeddings $\mathbf{z} = \mathbf{xE} + \mathbf{E}_{pos}$, where $\mathbf{E} \in \mathbb{R}^{(P^2 \cdot C) \times d}$ is a 2D-convolutional layer with kernel size and stride of size $P$, $d$ is the latent dimension size, and $\mathbf{E}_{pos} \in \mathbb{R}^{N \times d}$ are positional embeddings. Because inputs are linear sequences of patches rather than full 2D grids, we encode only horizontal (1D) positional information. Finally, the patch embeddings are processed through a stack of Transformer layers (Vaswani et al., 2017). A final linear layer projects the average over patch encodings from $d$ to the dimension of the language model input embeddings.

The fallback network is designed to function similarly to a vocabulary lookup, providing non-contextual embeddings which the language model can later contextualize. Specifically, we (1) pretokenize inputs into words,[1] (2) encode words independently of one another, and (3) apply average pooling over the patch encodings corresponding to a word to obtain a single word-level representation $\mathbf{y}_i \in \mathbb{R}^d$. Two key adjustments enable the efficient handling of multiple rendered words in a single forward pass: we concatenate the patches of individual words into a single sequence, resetting positional embeddings at each word boundary; and we restrict attention so that patches only attend to other patches within the same word.

**Text Compression**  Average-pooling the encoder representations leads to improved downstream efficiency by compressing subword-level information into a single embedding vector, shortening the input sequences provided to the language model. This advantage is particularly pronounced for non-Latin scripts prone to over-segmentation with an English-centric tokenizer. This compression effectively increases the amount of content that can fit within a language model's fixed context window.

**Interleaving Text and Image Representations**  The flexibility of our method allows words from the input text to be selectively embedded via the vocabulary or encoded as visual representations. For instance, non-Latin segments can be passed to the fallback network, while Latin (ASCII) segments go through the tokenizer. This selective encoding enables the language model to process only those parts of the input that align with its pretrained vocabulary, delegating more complex segments to the fallback network. We hypothesize that interleaving modalities within sentences is particularly advantageous for tasks involving *code-switching*, where a monolingual tokenizer may suboptimally represent parts of the input that the fallback network can be trained to handle.

## 3 Experiments with Decoder-only Models

To demonstrate the efficacy of our proposed fallback network, we focus on the task of machine translation from languages written in non-Latin scripts into English. Since English-centric models handle English generation reliably, this setup isolates the impact of improved input representation on the downstream task.

We conduct experiments using three decoder-only language models, namely SmolLM2-360M, SmolLM2-1.7B, and Phi-3-mini (3.8B parameters). These models are all based on the same underlying architecture (Touvron et al., 2023b) and finetuned for chat applications. SmolLM2 models have a vocabulary size of 49,152, whereas Phi-3-mini has 32,064 tokens.

---

[1] Splitting on whitespace is one simple *pretokenization* strategy; for languages without clear word boundaries, more appropriate segmentation methods can be utilized.

The linguistic capacity of all three models is mostly restricted to English text (Allal et al., 2025; Abdin et al., 2024). We follow the language models' default chat template.

### 3.1 Data and Experimental Setup

We train the models on parallel data from the OPUS corpus (Tiedemann, 2012) and evaluate them on the FLORES+ benchmark (NLLB Team et al., 2022). Specifically, we consider translations into English from Hindi (HI), Russian (RU), Spanish (ES), Thai (TH), and Ukrainian (UK).[2] Additional details are provided in Table 9 (Appendix A). Translation quality is measured using CHRF++ (Popović, 2015), a character $n$-gram $F$-score incorporating word unigrams and bigrams of the hypothesis with respect to the reference translation. CHRF++ is the standard primary metric for assessing performance on FLORES benchmarks (Goyal et al., 2022; NLLB Team et al., 2022; Costa-jussà et al., 2024).

We render input text as images using the PangoCairo rendering software,[3] segmenting each word into patches containing character bigrams, following Lotz et al. (2023). Based on preliminary experiments, we apply a sliding window with one-character overlap between patches, analogous to overlapping frames in speech modeling. For instance, the word *Happy* is segmented into patches of: $\boxed{\text{Ha}}$, $\boxed{\text{ap}}$, $\boxed{\text{pp}}$, and $\boxed{\text{py}}$.[4] We use the Google Noto font family for comprehensive script coverage.[5] Following Salesky et al. (2023), each patch is rendered as a $24 \times 24$ pixel image at 120 DPI with a font size of 10. The maximum sequence length of the fallback network is 529 image patches.

We constrain the fallback network to fewer than 100M parameters, approximately matching the embedding layer of SmolLM2-1.7B and Phi-3-mini. Based on preliminary experiments, we select a 92M-parameter configuration with $n_{\text{layers}} = 4$, $d_{\text{model}} = 1536$, and $n_{\text{heads}} = 16$. Section 3.6 explores alternative fallback network configurations.

Following the standard pretrain-then-finetune paradigm (Li et al., 2020), training proceeds in two stages: first, we pretrain the randomly initialized fallback network while freezing the language model, aligning the fallback network features to the language model (Peters et al., 2019; Kumar et al., 2022; Ren et al., 2023); next, we perform joint finetuning on the downstream task. During finetuning, we apply parameter-efficient updates using Weight-Decomposed Low-Rank Adaptation (DoRA; Liu et al., 2024a), employing reduced rank for the decoder and full rank for the fallback network. The learning rate is linearly warmed up to $3 \times 10^{-4}$ during the first 10% of training, followed by cosine decay to $3 \times 10^{-5}$. Additional experimental details are provided in Table 10 (Appendix A). Results for all experiments are averaged over three runs. Standard deviations are reported in Appendix B.

### 3.2 Competing Methods

We evaluate the pixel-based fallback network (PIXELS) against default model tokenization (BASE), vocabulary expansion (VOCAB+), and a byte-based fallback network (BYTES).

**Vocabulary Expansion**    To improve the language coverage of the language model, we train a new tokenizer and merge it into the original one, $\mathcal{V}_+ = \mathcal{V}_{\text{BASE}} \cup \mathcal{V}_{\text{new}}$. Specifically, we train another byte-level BPE tokenizer with a vocabulary size of 32k on either Hindi, Russian, or Thai. This results in expanded vocabulary sizes falling between the typical 30k-60k range of monolingual models (Brown et al., 2020; Touvron et al., 2023a) and the 100k+ token range of multilingual models (BigScience Workshop et al., 2023; Chowdhery et al., 2023; Dubey et al., 2024). This adds approximately 25M parameters to SmolLM2-360M, 50M parameters to SmolLM2-1.7B, and 90M parameters to Phi-3-mini. Following common practice, we randomly initialize the new vocabulary embeddings (Choi et al., 2024; Yamaguchi et al.,

---

[2]We word-tokenize Thai with DeepCut (Kittinaradorn et al., 2019) for fallback network modeling.
[3]https://docs.gtk.org/PangoCairo
[4]Not illustrated in Figure 1 for simplicity.
[5]https://fonts.google.com/noto

| | HI→EN | | | | RU→EN | | | | TH→EN | | | |
|---|---|---|---|---|---|---|---|---|---|---|---|---|
| | BASE | VOCAB+ | BYTES | PIXELS | BASE | VOCAB+ | BYTES | PIXELS | BASE | VOCAB+ | BYTES | PIXELS |
| SmolLM2-360M | 53.2 | 48.3 | 53.2 | **56.8** | 53.9 | 53.0 | 55.0 | **56.0** | 36.5 | 34.8 | 46.9 | **48.6** |
| SmolLM2-1.7B | 56.8 | 54.4 | 57.6 | **59.0** | 57.0 | 56.7 | 57.4 | **57.8** | 40.4 | 39.4 | 50.2 | **52.1** |
| Phi-3-mini | 57.3 | 54.7 | 59.5 | **60.9** | 57.9 | 57.8 | 57.8 | **58.2** | 51.1 | 50.4 | 52.0 | **53.1** |

Table 1: CHRF++ scores for XX→EN translation after finetuning for one epoch.

2024).[6] Training is done in two stages, with the new embeddings being pretrained in a first stage, followed by a stage of model finetuning, for a fair comparison to the fallback network.

**Byte-based Fallback Network**   Vocabulary-free modeling can alternatively be achieved by representing text at the byte level (Xue et al., 2022; Yu et al., 2023; Kallini et al., 2025), decomposing inputs into a discrete set of 256 embeddings. Unlike byte-level BPE, which uses byte sequences as subword units, treating text atomically as individual bytes enables complete vocabulary coverage without a large embedding matrix. However, byte-based modeling significantly increases sequence lengths, as each character may require multiple bytes depending on its Unicode encoding (Libovický et al., 2022). For instance, the source text shown in Figure 1 occupies six image patches but requires 59 bytes to represent. For byte-based fallback encoding, the maximum sequence length of the fallback network is therefore extended to 2048 bytes, significantly increasing GPU memory requirements.

To compare pixels to bytes as basis for vocabulary-free encoding, we train parallel fallback networks differing only in input modality and corresponding embedding layers.[7] Conceptually, this sets up a key trade-off for the fallback network: byte-level inputs yield longer sequences drawn from a discrete input space, whereas pixel-based inputs produce shorter sequences characterized by a continuous representation. This also quantifies the benefit to the language model derived from the added encoder capacity of the fallback network.

## 3.3   Machine Translation Results

Translation performances after one epoch of pretraining and finetuning are shown in Table 1. We observe that pixel-based representations (PIXELS) consistently outperform the other methods, including the byte-based fallback network (BYTES), with differences exceeding multiple run-to-run standard deviations (Table 14). Vocabulary expansion (VOCAB+) falls below even default tokenizer modeling (BASE), likely due to insufficient training to effectively integrate the newly added vocabulary tokens in this setup (Yamaguchi et al., 2024; Zhao et al., 2024). The SmolLM2-360M model particularly benefits from the fallback network, showing improvements ranging from 2 to 12 points. Notably, pixel-augmented SmolLM2-360M surpasses the larger SmolLM2-1.7B baseline on TH→EN (48.6 vs. 40.4), a trend also evident between SmolLM2-1.7B and Phi-3-mini (52.1 vs. 51.1).

## 3.4   Cross-lingual Transfer Results

To evaluate how effectively pixel-based representations facilitate positive language transfer (Conneau et al., 2020; Chau et al., 2020; Pfeiffer et al., 2021), particularly relevant for low-resource scenarios, we pretrain the fallback networks on 11M samples of RU→EN, ES→EN, or TH→EN, and subsequently finetune on UK→EN for $k$ steps, where the number of steps simulates constraints on available training data. As a comparison, we follow the same procedure for continued training of the language model embedding matrix. We compare performance to default modeling without continued embedding training (BASE*) and setups without fallback network pretraining (PIXELS*, BYTES*). We omit comparisons to vocabulary expansion due to its non-competitive effectiveness in Section 3.3.

---

[6]Mean instead of random initialization for vocabulary expansion does not change the conclusions.

[7]The embedding layer within the fallback network comprises 13M parameters for pixel-based encoding and 11M parameters for byte-based encoding.

| Steps | Only UK→EN | | | RU→EN then UK→EN | | | ES→EN then UK→EN | | | TH→EN then UK→EN | | |
|---|---|---|---|---|---|---|---|---|---|---|---|---|
| | BASE* | BYTES* | PIXELS* | BASE | BYTES | PIXELS | BASE | BYTES | PIXELS | BASE | BYTES | PIXELS |
| | | | | | | *SmolLM2-360M* | | | | | | |
| 10 | 18.8 | 11.7 | 13.3 | 21.1 | 25.6 | **31.2** | 18.9 | 15.0 | 14.6 | 19.9 | 14.6 | 13.5 |
| 50 | 23.3 | 12.9 | 13.4 | 24.5 | 34.2 | **40.2** | 23.3 | 16.8 | 20.9 | 23.5 | 16.8 | 18.0 |
| 100 | 26.0 | 15.4 | 15.2 | 26.8 | 39.2 | **44.4** | 25.9 | 19.3 | 29.8 | 25.9 | 18.6 | 25.0 |
| 1000 | 38.9 | 19.3 | 41.6 | 40.1 | 49.6 | **52.6** | 39.1 | 46.1 | 50.6 | 39.3 | 42.5 | 49.1 |
| | | | | | | *SmolLM2-1.7B* | | | | | | |
| 10 | 35.7 | 5.3 | 8.3 | **39.8** | 30.1 | 35.9 | 36.5 | 15.1 | 14.9 | 36.5 | 14.9 | 15.2 |
| 50 | 42.2 | 14.7 | 14.3 | 44.0 | 39.6 | **45.5** | 42.6 | 17.0 | 22.9 | 41.5 | 17.3 | 20.9 |
| 100 | 43.8 | 15.8 | 15.8 | 45.9 | 44.0 | **48.9** | 44.1 | 20.7 | 34.2 | 43.7 | 19.8 | 30.4 |
| 1000 | 51.2 | 27.0 | 46.9 | 52.1 | 53.2 | **55.7** | 51.1 | 48.9 | 53.2 | 51.5 | 46.7 | 52.4 |
| | | | | | | *Phi-3-mini* | | | | | | |
| 10 | 43.3 | 9.5 | 11.3 | **44.4** | 30.3 | 12.4 | 41.6 | 14.1 | 13.0 | 43.9 | 13.3 | 12.7 |
| 50 | 49.8 | 15.3 | 14.9 | 49.1 | 46.8 | **51.1** | 48.5 | 20.6 | 29.0 | 49.2 | 18.5 | 26.1 |
| 100 | 51.2 | 17.0 | 15.7 | 50.8 | 50.3 | **53.8** | 50.2 | 31.3 | 44.2 | 50.7 | 27.2 | 41.7 |
| 1000 | 56.6 | 36.1 | 54.5 | 56.6 | 57.5 | **58.8** | 55.8 | 55.4 | 57.3 | 56.1 | 54.0 | 56.9 |

Table 2: CHRF++ scores on UK→EN translation after *k* training steps, starting from weights initially trained on XX→EN. The "Only UK→EN" setting involves no prior training.

Table 2 shows that integrating a pixel-based fallback network generally yields the strongest transfer effects, particularly benefiting the SmolLM2-360M model. We attribute this improvement to the ViT's convolutional layer, which embeds inputs directly at the pixel level and enables updates to all encoder parameters at each training step. This promotes crosslingual transfer as the fallback network can exploit shared visual cues among languages (Rahman et al., 2023; Salesky et al., 2023), and most notably so with pretraining on Russian, which uses the same script as Ukrainian (Cyrillic.) Positive transfer for BYTES with Russian likely arises from the overlap in byte sequences encoding Cyrillic characters.

### 3.5 Cross-task Transfer Results

Beyond machine translation, we evaluate the potential of transfer across tasks by adapting a fallback network pretrained for HI→EN machine translation (from Section 3.3) to topic classification on the 10 languages from the SIB200 dataset (Adelani et al., 2024) written in the Devanagari script. Since pixel-based augmentation consistently outperformed the byte-based alternative in prior experiments, we now focus exclusively on PIXELS. See Table 11 (Appendix A) for experimental details.

Table 3 compares test set accuracies from finetuning the three language models with default tokenization (BASE) and with our fallback network (PIXELS). We find that augmenting Phi-3-mini results in reduced performance, potentially due to the fallback network overfitting during its machine translation pretraining. The SmolLM2 models, on the other hand, consistently benefit from the augmentation, especially so on the Hindi articles.

| | BASE | PIXELS |
|---|---|---|
| *SmolLM2-360M* | | |
| Hindi | 41.0 | **78.1** |
| Avg. Deva. | 40.1 | **65.1** |
| *SmolLM2-1.7B* | | |
| Hindi | 70.8 | **77.0** |
| Avg. Deva. | 70.0 | **72.2** |
| *Phi-3-mini* | | |
| Hindi | **72.5** | 70.3 |
| Avg. Deva. | **69.3** | 45.6 |

Table 3: Topic classification.

### 3.6 Efficiency Analysis

We observe that the relative computational overhead during training, introduced by the fallback network, varies with model scale and decreases for larger models (Table 4, based on experiments in Section 3.3). Although the first generation step incurs increased computational cost (measured in FLOPs), subsequent steps reuse cached fallback encodings. Crucially, for a similar number of generated tokens ("Gen len"), the shorter input sequences from fallback network compression significantly reduce total sequence-level inference time, particularly for Phi-3-mini and on Thai. On the FLORES+ dev set, the fallback network leads

|  | Train (s) | Gen (s) | Gen len | FLOPs |
|---|---|---|---|---|
| *SmolLM2 360M* | | | | |
| HI→EN | 1.74 | 0.96 | 0.97 | 1.41 |
| RU→EN | 1.76 | 0.98 | 0.98 | 1.41 |
| TH→EN | 1.75 | 0.61 | 0.88 | 1.41 |
| *SmolLM2 1.7B* | | | | |
| HI→EN | 1.42 | 0.92 | 1.00 | 1.09 |
| RU→EN | 1.43 | 0.97 | 1.00 | 1.09 |
| TH→EN | 1.42 | 0.68 | 0.93 | 1.09 |
| *Phi-3-mini* | | | | |
| HI→EN | 1.18 | 0.36 | 0.98 | 1.05 |
| RU→EN | 1.19 | 0.40 | 1.00 | 1.05 |
| TH→EN | 1.19 | 0.26 | 0.98 | 1.05 |

Table 4: Metric ratios (PIXELS/BASE).

| $n_{\text{params}}$ | $n_{\text{layers}}$ | $d_{\text{model}}$ | $n_{\text{heads}}$ | HI→EN |
|---|---|---|---|---|
| *SmolLM2 360M* | | | | |
| 92M | 4 | 1536 | 16 | 43.8 |
| 65M | 6 | 960 | 12 | 43.1 |
| 27M | 2 | 960 | 12 | 41.5 |
| *SmolLM2 1.7B* | | | | |
| 92M | 4 | 1536 | 16 | 51.8 |
| 51M | 4 | 1024 | 16 | 50.8 |
| 31M | 2 | 1024 | 16 | 50.1 |

Table 5: Fallback network configurations. Performance is measured as HI→EN translation quality after one epoch of pretraining when only updating the network parameters.

to average compression ratios for Hindi, Russian, and Thai of 5.1, 4.7, and 8.6, respectively, relative to the SmolLM2 tokenizer, and 5.1, 2.2, and 5.1 relative to the Phi-3-mini tokenizer.

To address the higher relative overhead incurred by the SmolLM2 models, we evaluate performance after machine translation pretraining on HI→EN for one epoch using scaled-down fallback network configurations (Table 5). Even at reduced capacity, the fallback networks largely retain their performance, indicating that the demonstrated benefits of pixel-augmented modeling are achievable at a reduced cost.

# 4   Interleaving Images and Text

The flexibility to interleave visual and textual representations is broadly relevant in multi-modal scenarios such as multi-image applications and visual storytelling (Li et al., 2025). To explore this flexibility within our proposed framework, we evaluate performance on a machine translation task involving Hindi-English code-switched source text and English target text from Tarunesh et al. (2021). When interleaving representations, ASCII text is embedded using the vocabulary, while all other segments are delegated to the HI→EN pretrained fallback network from Section 3.3. We compare the performance of interleaved modeling against default tokenization and uni-modal pixel processing, with which the entire input sequence is encoded by the fallback network. See Table 12 (Appendix A) for experimental details.

Table 6 shows that the fallback network again offers considerable gains over tokenization. Yet, mixing input modalities (PIXELS⬩) at best leads to the same performance as encoding the entire input via the fallback network (PIXELS). While the majority of the code-switched source text is indeed in Hindi (75%), this result raises questions about how compatible the two latent representation spaces are. Intuitively, handling English text via the tokenizer should be easier than having the fallback network learning a new language, especially given the limited amount of training data. We next explore this observation.

**Modality Gap**   We hypothesize that a disconnect between the latent spaces of images and text limits effective utilization of both modalities within a sequence. We therefore train a linear classifier on the FLORES+ dev set to distinguish Hindi words encoded by the HI→EN fallback network from English words embedded by the vocabulary. The classifier achieves perfect accuracy on a held-out subset, indicating fully disjoint latent spaces (Wang & Isola, 2020; Shi et al., 2023). Additionally, we measure the distance between the centers of these spaces (Liang et al., 2022), $||\mu_I - \mu_T||_2$. For SmolLM2-360M this distance is 40.7.

While it is unclear whether narrowing this gap would lead to better downstream performance (Al-Jaff, 2023; Yaras et al., 2024; Fahim et al., 2025), as the gap might arise from learning dynamics rather than representation quality, we propose new pretraining strategies aimed at better aligning image and text representations to facilitate effective mixed-modality

| | BASE | PIXELS◗ | PIXELS |
|---|---|---|---|
| SmolLM2-360M | 32.7 | **43.3** | **43.3** |
| SmolLM2-1.7B | 42.3 | **45.8** | **45.8** |
| Phi-3-mini | 44.9 | 45.9 | **47.8** |

Table 6: CHRF++ scores on Hindi-English code-switched data. "◗" indicates mixed input modality sequences.

| | $\|\|\mu_I - \mu_T\|\|_2$ | PIXELS◗ |
|---|---|---|
| SYNTHESIZED | 77.3 | 42.5 |
| PREFIX | 126.8 | 37.4 |
| ALIGNMENT | 2.6 | 38.4 |

Table 7: Distance between latent space centers and downstream performance on mixed-modality sequences. All experiments are based on SmolLM2-360M.

modeling: mixing input representations during pretraining of the fallback network and employing an auxiliary loss based on word alignments.[8]

**Pretraining on Modality-switched Data**  We explore two distinct pretraining strategies on the HI→EN machine translation data. (1) We obtain word alignments between source and target text in the HI→EN data and use those to synthesize code-switched data with the methodology outlined in Jalili Sabet et al. (2020), based on XLM-R$_{\text{LARGE}}$ (Conneau et al., 2020), matching the downstream Hindi-English ratio of 75:25 (SYNTHESIZED). (2) We extend the former approach by adding modality-indicating prefix tokens (Wang et al., 2024; Nguyen et al., 2025; Tschannen et al., 2025) to explicitly mark segment modality (PREFIX).

**Auxiliary Alignment Loss**  Related work has found explicit signals to aid the alignment of untied embedding spaces (Minixhofer et al., 2024). We therefore propose to include an auxiliary training objective during pretraining that forces the fallback network $h(w_k)$ to mimic the vocabulary embeddings $e_{w_k}$ for aligned words (Pinter et al., 2017)

$$\mathcal{L}^{\text{align}} = \frac{1}{n} \sum_{k=1}^{n} ||h(w_k) - e_{w_k}||_2^2.$$

Based on the word alignments from pretraining with modality-switched data, we combine $\mathcal{L}^{\text{align}}$ with the cross entropy loss $\mathcal{L}^{\text{CE}}$ to obtain the new loss (ALIGNMENT).

$$\mathcal{L} = \mathcal{L}^{\text{CE}} + \mathcal{L}^{\text{align}}.$$

**Results Using Alignment Strategies**  Table 7 shows that none of the proposed strategies outperform the baseline from Table 6 (43.3). In all settings, we again find that a linear classifier can perfectly separate the two modalities. Notably, pretraining and finetuning with the auxiliary loss (ALIGNMENT) reduces the distance between centers (2.6 vs. 40.7) but leads to substantially worse performance. These findings indicate that neither simple alignment strategies nor reducing latent-space distance alone effectively improves performance or bridges the latent spaces. Future work could explore more sophisticated methods for effectively interleaving text and image representations.

## 5  Experiments with Encoder-only Models

To explore whether the benefits of a pixel-based fallback network generalize to different architectures, we experiment with BERT (Devlin et al., 2019), which unlike BPE-based models suffers from out-of-vocabulary constraints on unseen scripts (Rust et al., 2021). Bypassing the tokenizer with a fallback network avoids potential [UNK] token substitution and thereby loss of information. Specifically, we augment BERT$_{\text{BASE}}$ with a 24M-parameter pixel-based fallback network.[9] We evaluate on named entity recognition in Indic languages from the

---

[8]All fallback networks in this section share the same initialization, as initial randomness could affect the representation space (Liang et al., 2022).

[9]$n_{\text{layers}} = 4$, $d_{\text{model}} = 768$, and $n_{\text{heads}} = 12$.

| | $\lvert\theta\rvert$ | BN | GU | HI | KN | ML | MR | OR | PA | TA | TE | Avg. |
|---|---|---|---|---|---|---|---|---|---|---|---|---|
| mBERT$_{\text{BASE}}$ | 179M | 77.5 | 78.7 | 79.7 | 76.5 | 78.6 | 79.1 | 23.8 | 68.1 | 67.5 | 79.5 | 70.9 |
| BERT$_{\text{BASE}}$ | 110M | 62.2 | 24.3 | 62.5 | 25.7 | 32.0 | 65.7 | 23.8 | 13.1 | 15.2 | 26.8 | 35.1 |
| BERT+PIXELS* | 134M | **69.8** | **73.5** | **74.9** | 71.1 | 71.0 | **76.5** | 24.6 | **65.8** | 51.6 | **73.1** | **65.2** |
| BERT+PIXELS | 134M | 66.8 | 72.7 | – | **72.4** | **72.8** | 75.3 | **26.4** | 63.7 | **57.3** | 71.8 | 64.4 |
| BERT$_{\text{LARGE}}$ | 340M | 62.6 | 24.3 | 63.7 | 25.6 | 31.8 | 66.5 | 22.7 | 13.6 | 15.3 | 25.8 | 35.2 |
| BERT [UNK]% | | 9.4% | 85.6% | 14.8% | 81.0% | 79.5% | 11.4% | 85.8% | 85.4% | 62.7% | 80.6% | 59.6% |
| mBERT [UNK]% | | 0.0% | 0.0% | 0.0% | 0.0% | 0.0% | 0.0% | 85.8% | 0.2% | 0.0% | 0.0% | 8.6% |

Table 8: Test set $F_1$ scores for BERT models on Naamapadam. $\lvert\theta\rvert$ denotes parameter count. The bottom two rows report the proportion of [UNK] tokens for BERT and mBERT.

Naamapadam dataset (Mhaske et al., 2023),[10] a semantic sequence-level classification task. The models are fully finetuned, encoding the entire input via the fallback network. We compare performance with a randomly initialized fallback network (BERT+PIXELS*) and after pretraining on the Hindi portion of the dataset (BERT+PIXELS).

Table 8 shows that integrating a fallback network substantially alleviates BERT's representational limitations, outperforming the equally constrained BERT$_{\text{LARGE}}$. For these tasks, pretraining the fallback network provides no additional benefit, likely because finetuning on enough data sufficiently adapts these smaller models to a comparatively simpler task than open-ended text generation (Liang et al., 2023). However, BERT+PIXELS*, while competitive, does not surpass the multilingual mBERT, which was pretrained on 104 languages. We observe a significant correlation between the proportion of [UNK] tokens and the gap in performance between BERT and BERT+PIXELS*.[11] These findings reinforce that pixel-based fallback networks provide an effective approach to overcoming the vocabulary constraints of monolingual models in multilingual scenarios.

## 6    Related Work

In multilingual modeling, computational constraints often prohibit adequately representing a large number of languages (Conneau et al., 2020; Rust et al., 2021). Such vocabulary constraints result in lower downstream performance for languages underrepresented during pretraining (Bostrom & Durrett, 2020; Toraman et al., 2023; Fujii et al., 2023). Recent approaches to vocabulary-free NLP typically fall into one of two categories: byte-based or pixel-based methods.

While overlapping byte sequences are not necessarily semantically related (Choi et al., 2024; Cui et al., 2024), shared sequences can enhance robustness and facilitate cross-lingual transfer via parameter sharing (Xue et al., 2022). De Souza et al. (2024) rely on bytes for quantifying also the language-agnostic component to cross-lingual transfer. To alleviate the overhead from modeling non-Latin characters as bytes (Arnett et al., 2024), patch-based and dynamic token-merging strategies can improve the computational efficiency (Yu et al., 2023; Kallini et al., 2024). As a promising outlook, Byte Latent Transformer (Pagnoni et al., 2024) and EvaByte (Zheng et al., 2025) demonstrate comparable performance to subword LLMs.

Recent advances in pixel-based language modeling have demonstrated visual language understanding through pixels alone (Lee et al., 2023). In a similar vein, CLIPPO shows that a single encoder can effectively handle both image and text modalities by treating text as pixels (Tschannen et al., 2023). Our work builds upon the concept of a general-purpose pixel-based language encoder introduced in PIXEL (Rust et al., 2023). Lotz et al. (2023) further explored text rendering strategies for PIXEL to reduce input redundancy, while recent efforts by Chai et al. (2024) and Tai et al. (2024) investigated autoregressive pretraining directly on pixel representations, with Chai et al. (2024) finding benefits to multimodal over unimodal (text or image) pretraining. Additionally, Salesky et al. (2021; 2023) trained encoder-decoder

---

[10]We exclude Assamese since its run-to-run variance across all models exceeds that of the other languages by more than an order of magnitude.

[11]Pearson correlation $r = 0.67$, $p < 0.05$.

models for machine translation using pixels as inputs. In contrast, our approach enables pretrained and post-trained language models to benefit from pixel-based modeling without altering the underlying language model weights.

# 7 Conclusion

We introduced a fallback network that alleviates the vocabulary constraints of monolingual language models in multilingual settings by encoding text as pixels. Our experiments show that pixel-based encodings outperform default tokenization, standard vocabulary expansion, and byte-based methods, resulting in improved performance, shorter input sequences, and faster decoding compared to modeling without a fallback network. Notably, a pixel-augmented 360M-parameter model can surpass an unmodified 1.7B-parameter baseline on machine translation. Our fallback network also enables effective cross-task transfer, and cross-lingual transfer based on visual similarities between scripts. Interleaving text and image representations is an exciting direction and future work could explore more sophisticated methods for effectively and seamlessly mixing modalities within a sequence.

While our study focuses on English-centric models, we argue that our approach remains relevant also for multilingual models. Any model relying on a single, fixed-size vocabulary is faced with a trade-off between its vocabulary coverage and the granularity of its representations. The pixel-based fallback network is designed to circumvent the issue of over-segmentation, moving beyond vocabulary constraints instead of optimizing within them.

## Acknowledgements

We thank Matthias Sperber, Sarthak Garg, Desmond Elliott, Elizabeth Salesky, Phillip Rust, Kentaro Inui, Benjamin Heinzerling, and Ana Brassard for helpful discussions and feedback. Jonas F. Lotz is funded by the ROCKWOOL Foundation (grant 1242).

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

# A   Training Details

| Language | ISO 639-1 | Language Family | Script |
|---|---|---|---|
| Bengali | BN | Indo-Aryan | Bengali |
| English | EN | Indo-European | Latin |
| Gujarati | GU | Indo-European | Gujarati |
| Hindi | HI | Indo-European | Devanagari |
| Kannada | KN | Dravidian | Kannada |
| Malayalam | ML | Dravidian | Malayalam |
| Marathi | MR | Indo-European | Devanagari |
| Oriya | OR | Indo-European | Oriya |
| Punjabi | PA | Indo-European | Gurmukhi |
| Russian | RU | Indo-European | Cyrillic |
| Spanish | ES | Indo-European | Latin |
| Tamil | TA | Dravidian | Tamil |
| Telugu | TE | Dravidian | Telugu |
| Thai | TH | Kra-Dai | Thai |
| Ukrainian | UK | Indo-European | Cyrillic |

Table 9: Overview of languages used in our experiments.

| Parameter | Value |
|---|---|
| Optimizer | AdamW (Loshchilov & Hutter, 2019; Kingma & Ba, 2015) |
| Adam $\beta$ | (0.9; 0.999) |
| Adam $\epsilon$ | $1 \times 10^{-8}$ |
| Weight decay | 0.0 |
| Dropout probability | 0.0 |
| Maximum source length | 256 |
| Maximum target length | 256 |
| Learning rate schedule | Cosine Decay (Loshchilov & Hutter, 2017) |
| Warmup ratio | 10% |
| Peak learning rate | $3 \times 10^{-4}$ |
| Minimum learning rate | $3 \times 10^{-5}$ |
| Batch size | SmolLM2: 256; Phi-3-mini: 512 |
| Number of training samples in 1 epoch | Hindi: 14M, Russian: 14M, Spanish: 14M, Thai: 11M |
| (DoRA) Rank $r$ | 32 |
| (DoRA) $\alpha$ | 64 |
| (DoRA) dropout | 0.05 |
| (DoRA) Modules | Q, K, V, O and fallback network or LM embedding matrix |
| Beam size | 2 |
| Length penalty | 1.0 |
| Repetition penalty | 1.0 |
| Temperature | 1.0 |
| Top-K sampling | 50 |
| Top-P sampling | 1.0 |

Table 10: Parameters and their values for the machine translation experiments in Section 3.3 and 3.4. The top section covers training and the bottom covers inference.

Pretrained language model weights are downloaded from Hugging Face.[12,13,14]

---

[12]https://huggingface.co/HuggingFaceTB/SmolLM2-360M-Instruct

[13]https://huggingface.co/HuggingFaceTB/SmolLM2-1.7B-Instruct

[14]https://huggingface.co/microsoft/Phi-3-mini-4k-instruct

| Parameter | Value |
|---|---|
| Batch size | 64 |
| Max number of epochs | 10 |
| Early stopping | ✓ |

Table 11: Parameters and their values for the topic classification experiments in Section 3.5. Only the batch size and and number of epochs are different from the experiments in Section 3.3 and 3.4. We apply early stopping to check for convergence before the maximum number of epochs. We instruct the models using the template: `Would you classify the topic of this article as "science/technology", "travel", "politics", "sports", "health", "entertainment", or "geography"? {INPUT}`.

| Parameter | Value |
|---|---|
| Batch size | 64 |
| Epochs | 2 (342 steps) |

Table 12: Parameters and their values for the code-switching experiments in Section 4. Only the batch size and and number of epochs are different from the experiments in Section 3.3 and 3.4.

| Parameter | Value |
|---|---|
| Optimizer | AdamW |
| Adam $\beta$ | (0.9; 0.999) |
| Adam $\epsilon$ | $1 \times 10^{-8}$ |
| Weight decay | 0.0 |
| DoRA dropout | 0.05 |
| Maximum sequence length | 192 |
| Learning rate schedule | Linear Decay |
| Warmup steps | 1000 |
| Learning rate | $3 \times 10^{-4}$ |
| Batch size | 64 |
| Max number of training samples | 100,000 |
| Max steps | 15,000 |
| Eval steps | 500 |
| Early stopping | ✓ |

Table 13: Parameters and their values for the NER experiments in Section 5.

# B Detailed Experimental Results

Standard deviations are reported using subscript notation.

| | HI→EN | | | | RU→EN | | | | TH→EN | | | |
|---|---|---|---|---|---|---|---|---|---|---|---|---|
| | BASE | VOCAB+ | BYTES | PIXELS | BASE | VOCAB+ | BYTES | PIXELS | BASE | VOCAB+ | BYTES | PIXELS |
| SmolLM2-360M | $53.2_{0.36}$ | $48.3_{0.26}$ | $53.2_{0.13}$ | $\mathbf{56.8}_{0.49}$ | $53.9_{0.12}$ | $53.0_{0.17}$ | $55.0_{0.12}$ | $\mathbf{56.0}_{0.18}$ | $36.5_{0.22}$ | $34.8_{0.05}$ | $46.9_{0.41}$ | $\mathbf{48.6}_{0.18}$ |
| SmolLM2-1.7B | $56.8_{0.15}$ | $54.4_{0.41}$ | $57.6_{0.08}$ | $\mathbf{59.0}_{0.10}$ | $57.0_{0.13}$ | $56.7_{0.17}$ | $57.4_{0.08}$ | $\mathbf{57.8}_{0.09}$ | $40.4_{0.18}$ | $39.4_{0.04}$ | $50.2_{0.10}$ | $\mathbf{52.1}_{0.16}$ |
| Phi-3-mini | $57.3_{0.14}$ | $54.7_{0.22}$ | $59.5_{0.13}$ | $\mathbf{60.9}_{0.20}$ | $57.9_{0.13}$ | $57.8_{0.03}$ | $57.8_{0.11}$ | $\mathbf{58.2}_{0.12}$ | $51.1_{0.26}$ | $50.4_{0.32}$ | $52.0_{0.37}$ | $\mathbf{53.1}_{0.35}$ |

Table 14: Copy of Table 1 including standard deviations.

| | Only UK→EN | | | RU→EN then UK→EN | | | ES→EN then UK→EN | | | TH→EN then UK→EN | | |
|---|---|---|---|---|---|---|---|---|---|---|---|---|
| Steps | BASE | BYTES* | PIXELS* | BASE | BYTES | PIXELS | BASE | BYTES | PIXELS | BASE | BYTES | PIXELS |
| | | | | | | *SmolLM2-360M* | | | | | | |
| 10 | $18.8_{0.18}$ | $11.7_{1.61}$ | $13.3_{0.25}$ | $21.1_{0.23}$ | $25.6_{0.16}$ | $\mathbf{31.2}_{0.18}$ | $18.9_{0.63}$ | $15.0_{0.05}$ | $14.6_{0.16}$ | $19.9_{0.16}$ | $14.6_{0.21}$ | $13.5_{0.21}$ |
| 50 | $23.3_{0.14}$ | $12.9_{0.36}$ | $13.4_{0.35}$ | $24.5_{0.29}$ | $34.2_{0.10}$ | $\mathbf{40.2}_{0.17}$ | $23.3_{0.18}$ | $16.8_{0.11}$ | $20.9_{0.06}$ | $23.5_{0.03}$ | $16.8_{0.13}$ | $18.0_{0.09}$ |
| 100 | $26.0_{0.15}$ | $15.4_{0.20}$ | $15.2_{0.11}$ | $26.8_{0.09}$ | $39.2_{0.06}$ | $\mathbf{44.4}_{0.07}$ | $25.9_{0.14}$ | $19.3_{0.11}$ | $29.8_{0.07}$ | $25.9_{0.18}$ | $18.6_{0.11}$ | $25.0_{0.25}$ |
| 1000 | $38.9_{0.16}$ | $19.3_{0.13}$ | $41.6_{0.91}$ | $40.1_{0.15}$ | $49.6_{0.08}$ | $\mathbf{52.6}_{0.08}$ | $39.1_{0.46}$ | $46.1_{0.38}$ | $50.6_{0.18}$ | $39.3_{0.50}$ | $42.5_{0.32}$ | $49.1_{0.32}$ |
| | | | | | | *SmolLM2-1.7B* | | | | | | |
| 10 | $35.7_{0.31}$ | $5.3_{1.29}$ | $8.3_{0.31}$ | $\mathbf{39.8}_{0.28}$ | $30.1_{0.13}$ | $35.9_{0.11}$ | $36.5_{0.37}$ | $15.1_{0.22}$ | $14.9_{0.09}$ | $36.5_{0.20}$ | $14.9_{0.13}$ | $15.2_{0.17}$ |
| 50 | $42.2_{0.25}$ | $14.7_{0.28}$ | $14.3_{0.60}$ | $44.0_{0.37}$ | $39.6_{0.29}$ | $\mathbf{45.5}_{0.11}$ | $42.6_{0.31}$ | $17.0_{0.03}$ | $22.9_{0.22}$ | $41.5_{0.01}$ | $17.3_{0.06}$ | $20.9_{0.03}$ |
| 100 | $43.8_{0.26}$ | $15.8_{0.27}$ | $15.8_{0.29}$ | $45.9_{0.07}$ | $44.0_{0.10}$ | $\mathbf{48.9}_{0.13}$ | $44.1_{0.42}$ | $20.7_{0.36}$ | $34.2_{0.10}$ | $43.7_{0.48}$ | $19.8_{0.18}$ | $30.4_{0.13}$ |
| 1000 | $51.2_{0.27}$ | $27.0_{0.26}$ | $46.9_{0.17}$ | $52.1_{0.18}$ | $53.2_{0.40}$ | $\mathbf{55.7}_{0.15}$ | $51.1_{0.34}$ | $48.9_{0.03}$ | $53.2_{0.13}$ | $51.5_{0.32}$ | $46.7_{0.07}$ | $52.4_{0.12}$ |
| | | | | | | *Phi-3-mini* | | | | | | |
| 10 | $43.3_{0.04}$ | $9.5_{0.57}$ | $11.3_{0.54}$ | $\mathbf{44.4}_{0.25}$ | $30.3_{1.01}$ | $12.4_{0.98}$ | $41.6_{0.02}$ | $14.1_{0.33}$ | $13.0_{0.54}$ | $43.9_{0.41}$ | $13.3_{0.40}$ | $12.7_{0.50}$ |
| 50 | $49.8_{0.16}$ | $15.3_{0.05}$ | $14.9_{0.08}$ | $49.1_{0.42}$ | $46.8_{0.34}$ | $\mathbf{51.1}_{0.29}$ | $48.5_{0.33}$ | $20.6_{0.23}$ | $29.0_{0.96}$ | $49.2_{0.09}$ | $18.5_{0.18}$ | $26.1_{0.29}$ |
| 100 | $51.2_{0.12}$ | $17.0_{0.09}$ | $15.7_{0.56}$ | $50.8_{0.28}$ | $50.3_{0.33}$ | $\mathbf{53.8}_{0.29}$ | $50.2_{0.16}$ | $31.3_{0.21}$ | $44.2_{0.24}$ | $50.7_{0.16}$ | $27.2_{1.09}$ | $41.7_{0.06}$ |
| 1000 | $56.6_{0.17}$ | $36.1_{0.52}$ | $54.5_{0.09}$ | $56.6_{0.03}$ | $57.5_{0.13}$ | $\mathbf{58.8}_{0.21}$ | $55.8_{0.15}$ | $55.4_{0.16}$ | $57.3_{0.16}$ | $56.1_{0.21}$ | $54.0_{0.14}$ | $56.9_{0.15}$ |

Table 15: Copy of Table 2 including standard deviations.

| | BASE | PIXELS |
|---|---|---|
| *SmolLM2-360M* | | |
| Hindi | $41.0_{2.32}$ | $\mathbf{78.1}_{3.19}$ |
| Avg. Deva. | 40.1 | **65.1** |
| *SmolLM2-1.7B* | | |
| Hindi | $70.8_{0.75}$ | $\mathbf{77.0}_{1.30}$ |
| Avg. Deva. | 70.0 | **72.2** |
| *Phi-3-mini* | | |
| Hindi | $\mathbf{72.5}_{1.30}$ | $70.3_{1.72}$ |
| Avg. Deva. | **69.3** | 45.6 |

Table 16: Copy of Table 3 including standard deviation.

| | BASE | PIXELS◐ | PIXELS |
|---|---|---|---|
| SmolLM2-360M | $32.7_{0.06}$ | $\mathbf{43.3}_{0.08}$ | $\mathbf{43.3}_{0.22}$ |
| SmolLM2-1.7B | $42.3_{0.09}$ | $\mathbf{45.8}_{0.24}$ | $\mathbf{45.8}_{0.33}$ |
| Phi-3-mini | $44.9_{0.10}$ | $45.9_{0.17}$ | $\mathbf{47.8}_{0.17}$ |

Table 17: Copy of Table 6 including standard deviations.

| | $||\mu_I - \mu_T||_2$ | PIXELS◐ |
|---|---|---|
| SYNTHESIZED | 77.3 | $42.5_{0.37}$ |
| PREFIX | 126.8 | $37.4_{0.02}$ |
| ALIGNMENT | 2.6 | $38.4_{0.16}$ |

Table 18: Copy of Table 7 including standard deviations.

| | $|\theta|$ | BN | GU | HI | KN | ML | MR | OR | PA | TA | TE | Avg. |
|---|---|---|---|---|---|---|---|---|---|---|---|---|
| mBERT$_{\text{BASE}}$ | 179M | $77.5_{1.12}$ | $78.7_{0.74}$ | $79.7_{1.02}$ | $76.5_{1.27}$ | $78.6_{0.16}$ | $79.1_{0.77}$ | $23.8_{2.34}$ | $68.1_{0.50}$ | $67.5_{0.10}$ | $79.5_{0.76}$ | 70.9 |
| BERT$_{\text{BASE}}$ | 110M | $62.2_{0.42}$ | $24.3_{0.70}$ | $62.5_{0.56}$ | $25.7_{1.31}$ | $32.0_{0.57}$ | $65.7_{0.63}$ | $23.8_{2.36}$ | $13.1_{0.62}$ | $15.2_{0.88}$ | $26.8_{0.32}$ | 35.1 |
| BERT+24M* | 134M | $\mathbf{69.8}_{1.01}$ | $\mathbf{73.5}_{1.13}$ | $\mathbf{74.9}_{0.10}$ | $71.1_{1.33}$ | $71.0_{1.25}$ | $\mathbf{76.5}_{0.32}$ | $24.6_{2.44}$ | $\mathbf{65.8}_{0.59}$ | $51.6_{2.20}$ | $\mathbf{73.1}_{2.74}$ | $\mathbf{65.2}$ |
| BERT+24M | 134M | $66.8_{1.01}$ | $72.7_{0.60}$ | – | $\mathbf{72.4}_{0.09}$ | $\mathbf{72.8}_{0.72}$ | $75.3_{0.86}$ | $\mathbf{26.4}_{1.00}$ | $63.7_{0.88}$ | $\mathbf{57.3}_{0.15}$ | $71.8_{0.62}$ | 64.4 |
| BERT$_{\text{LARGE}}$ | 340M | $62.6_{0.60}$ | $24.3_{0.79}$ | $63.7_{0.43}$ | $25.6_{1.67}$ | $31.8_{0.43}$ | $66.5_{1.65}$ | $22.7_{0.41}$ | $13.6_{0.24}$ | $15.3_{0.68}$ | $25.8_{0.06}$ | 35.2 |
| BERT [UNK]% | | 9.4% | 85.6% | 14.8% | 81.0% | 79.5% | 11.4% | 85.8% | 85.4% | 62.7% | 80.6% | 59.6% |
| mBERT [UNK]% | | 0.0% | 0.0% | 0.0% | 0.0% | 0.0% | 0.0% | 85.8% | 0.2% | 0.0% | 0.0% | 8.6% |

Table 19: Copy of Table 8 including standard deviations.

