# OpenReview forum: "Overcoming Vocabulary Constraints with Pixel-level Fallback"
_colmweb.org/COLM/2025/Conference — COLM 2025_

### Official Review · Reviewer_KQmA · 2025-04-21

**Rating:** 6
**Confidence:** 4
**Ethics Flag:** 1

**Summary:**

This paper proposes a pixel-level fallback method for enhanced input representation in a cross-lingual setup, motivated by the fact that LLMs tend to be English-centric and often show suboptimal performance in non-English languages. The results mainly on target-to-English machine translation tasks demonstrate that the proposed method generally outperforms base, vocabulary expansion, and byte-level models in terms of task performance. Also, it shows better inference efficiency over the base model.

**Questions To Authors:**

While not a substantial issue, why does this paper mainly focus on machine translation tasks despite the fact that it uses chat models? I think chat models should be evaluated on chat-related tasks like (multilingual) MT-Bench, MGSM, and multilingual IFEval etc.

**Reasons To Accept:**

1. This paper proposes a novel method of using a pixel network (encoder) to compress and improve input text representations that can suffer from suboptimal tokenization.
2. This paper conducts extensive experiments using both encoder-based and decoder-based models to verify the effectiveness of the proposed pixel-level fallback method.
3. The paper further supports the efficacy of integrating the pixel-level fallback method into the existing text-based tokenization with a detailed ablation analysis on interleaving images and text.

**Reasons To Reject:**

1. **Lack of baseline**
This work appears closely related to Zero-Shot Tokenizer Transfer (ZeTT) (Minixhofer et al., NeurIPS 2024) in its utilization of embeddings from an auxiliary network to mitigate the challenges of suboptimal tokenization for languages other than English. However, ZeTT is not compared in the paper. How does the performance of this work compare to ZeTT?

2. **Potentially limited applicability of the approach to different types of tasks**
Unlike ZeTT and vocabulary expansion, the proposed approach is only applicable to input embeddings; and no text compression benefit comes from an LM head. I wonder whether the proposed approach works well in generative target language tasks (i.e. non cross-lingual tasks like summarization (e.g. XLSUM) and reasoning (e.g. MGSM) tasks.)

3. **Unfair experimental setup for vocabulary expansion**
The efficacy of vocabulary expansion, especially in low-resource settings like this study, is likely underestimated by random initialization. Given that mean initialization has become the prevalent method in recent studies (Fujii et al., COLM 2024; Tejaswi et al., EMNLP Findings 2024; Mundra et al., CoNLL 2024) due to its significantly better performance than random initialization, how does the proposed approach perform against vocabulary expansion with mean initialization?

    (Fujii et al., COLM 2024) Continual Pre-Training for Cross-Lingual LLM Adaptation: Enhancing Japanese Language Capabilities
    (Tejaswi et al., EMNLP Findings 2024) Exploring Design Choices for Building Language-Specific LLMs
    (Mundra et al., CoNLL 2024) An Empirical Comparison of Vocabulary Expansion and Initialization Approaches For Language Models

4. **Potentially limited usefulness over recent LLMs**
This paper uses SmolLM2 and Phi-3 in their experiments, which are all English-centric. However, recent popular LLMs like Gemma2/3 and Llama 3.x are all multilingual and have far larger vocabulary sizes (e.g. 128K for Llama 3.x and 256K for Gemma2). This can potentially diminish the benefits of the proposed compression method in terms of both task performance and inference efficiency, as they can encode non-English texts far more efficiently than the models tested in this paper. Does the proposed method perform effectively with such popular models?

---

> ### Author Response · Authors · 2025-06-02
> **Reply to Reviewer KQmA**
>
> We thank the reviewer for their time and feedback. We address the concerns below.
>
> > Zero-Shot Tokenizer Transfer (ZeTT) (Minixhofer et al., NeurIPS 2024) not included as baseline
>
> We acknowledge the conceptual similarity to ZeTT, as both methods utilize a separate module to generate embeddings.
> However, their goals, mechanisms and empirical focus differ so fundamentally that we deem the comparison to be unwarranted.
>
> **Vocabulary dependence** Our work is vocabulary-free, completely bypassing vocabulary constraints. In contrast, ZeTT operates within these constraints and instead swaps the tokenizer for every new domain.
> This either assumes the existence of suitable tokenizers or requires training new ones for all intended purposes.
> Our work is motivated by moving *beyond* vocabulary constraints, not optimizing within them.
>
> **Training** Our method adds minimal overhead compared to ZeTT. For example, training a hypernetwork for Llama 3 takes around 4 days on a TPU-4-32 pod (https://github.com/bminixhofer/zett/issues/1#issuecomment-2120752565).
> Training and evaluating new tokenizers for swapping adds another layer of complexity to ZeTT.
>
> **Performance** ZeTT improves efficiency while preserving performance. Our method is shown to substantially boost performance, enabling smaller models to outperform larger ones, while also improving efficiency through input compression.
>
> We will include a discussion of these distinctions in the revised version of the paper.
>
> > No generative tasks in target language
>
> The primary goal of this work is to measure the impact of improved input representations, especially for languages underrepresented by a model's tokenizer.
> Focusing on translation into English provides a controlled setup where improvements can be traced unambiguously to input representations.
> We do, however, demonstrate the versatility of our approach on non-generative tasks like topic classification and NER.
>
> We agree that non-English generation is important and see augmentations of the LM head as an interesting direction for future work.
>
> > Mean initialization vs random initialization for vocabulary expansion
>
> Clarification: The text embeddings are initially trained for one epoch, updating only the embeddings, to establish a stronger starting point for finetuning (L164-165).
> This is in line with the setup in Dobler & Melo (2023), who additionally find that the difference between random initialization and even more advanced methods diminishes as training progresses.
> Furthermore, Salesky et al. (2020) demonstrate no clear advantage between random and mean embedding initializations in their machine translation experiments.
>
> In line with these previous findings, we repeated the vocabulary expansion baseline for SmolLM2-360M with mean-initialised embeddings and found conclusions remain unchanged. Mean vs random: Hi->En 47.1 vs 48.3, Ru->En 52.7 vs 53.0, Th->En 36.5 vs 34.8.
> Results for SmolLM2-1.7B and Phi-3-mini to follow.
>
> Konstantin Dobler & Gerard de Melo. 2023. FOCUS: Effective Embedding Initialization for Monolingual Specialization of Multilingual Models. EMNLP.
>
> Elizabeth Salesky et al. 2020. Optimizing Segmentation Granularity for Neural Machine Translation. Machine Translation.
>
> > Comparison to multilingual LLMs
>
> While today’s multilingual LLMs have much larger token inventories, a fixed vocabulary still imposes constraints and the proposed method remains  relevant for several reasons:
>
> **Open-Vocabulary** No finite subword vocabulary, regardless of its size, can effectively cover every possible script, specialized symbol, or future evolution of language.
> Our pixel-based method provides an egalitarian fallback to handle any character or script that can be digitally typeset without backing off to bytes.
>
> **Cross-lingual transfer** A large vocabulary does not guarantee high performance on languages that were not well-represented in the pretraining data. Our experiments demonstrate that a pixel-based encoder substantially improves performance for unseen languages by facilitating cross-lingual transfer between visually similar scripts. This makes our approach a valuable and efficient method for adapting models to new or low-resource languages.
>
> **Input compression** By mapping each word to a single vector, the fallback network shortens input sequences even for large vocabularies when the tokenizer fertility is > 1.  This lowers the KV-cache usage and speeds up inference.
>
> We will further discuss the trade-off of large multilingual vocabularies versus smaller pixel-augmented models in the revised version of the paper.
>
> > Why machine translation tasks with instruction-tuned models?
>
> We used instruction-tuned models as they represent the current state-of-the-art for publicly available LLMs at this scale. This demonstrates the potential of our approach when augmenting models that are actively valuable and relevant.
>
> Please let us know if further clarifications are needed.

---

> > ### Comment · Reviewer_KQmA · 2025-06-03
> >
> > Thanks for the response and for supplying some results for the mean initialization setup.
> >
> > I've carefully considered the points, but my concerns regarding the paper's applicability and the discussion on recent LLMs persist. Therefore, I'll keep my current score as is.
> >
> > Here's why:
> > (i) I still think that the applicability/usefulness of the proposed approach seems limited.  When used with target-to-target generation tasks, as mentioned in the review, does it perform well?  Concrete results in this setup would significantly strengthen the paper.
> > (ii) The discussion on recent LLMs isn't convincing without accompanying concrete results. In lines 51-60, the paper emphasizes benefits in both performance improvement and inference efficiency. Do these advantages hold for recent, large-vocabulary frontier models?
> > (iii) Finally, I find the reliance on two papers that did not experiment with decoder-based LMs to back the discussion on the effect of mean initialization unconvincing. This specific point requires support from more relevant papers that use decoder-based LMs.

---

> > > ### Author Response · Authors · 2025-06-08
> > >
> > > We thank the reviewer for the opportunity to clarify and strengthen our paper with additional results.
> > >
> > > > (i) I still think that the applicability/usefulness of the proposed approach seems limited. When used with target-to-target generation tasks, as mentioned in the review, does it perform well? Concrete results in this setup would significantly strengthen the paper.
> > >
> > > We thank the reviewer for highlighting concrete monolingual summarization (xx→xx) experiments that could strengthen the paper. As shown in Table R.1, our pixel-augmented approach consistently improves performance over the baseline.
> > >
> > > Experimental Setup: We finetuned the SmolLM models on monolingual summarization for Hindi, Russian, and Thai using data from XL-Sum (Hasan et al., 2021). Models were trained for ~1000 steps with a batch size of 64 (1 epoch for Hindi and Russian, 10 epochs for Thai). We compare our pixel-augmented model ("pixels") against the baseline model ("base"), which was initially trained on translation (Section 3).
> > >
> > > We would like to point out that SmolLM2 uses **tied input-output embeddings**, giving the baseline a strong starting point for generation since it has already learned good representations during the initial training (Pappas et al., 2018; Bertolotti & Cazzola, 2024). Importantly, the pixel-augmented model has not seen these initial updates to its input-output embeddings. Still, we find that our proposed approach improves non-English generation.
> > >
> > > We hope these results address the reviewer’s concern about the broader applicability of our method.
> > >
> > > |Method|Hindi|Russian|Thai|
> > > |-|:-:|:-:|:-:|
> > > | | | **SmolLM-2 360M** | |
> > > |base|19.8|18.7|15.4|
> > > |pixels|21.9|19.6|15.7|
> > > | | |**SmolLM-2 1.7B**| |
> > > |base|21.7|20.1|16.6|
> > > |pixels|24.2|21.1|17.0|
> > >
> > > Table R.1: chrF++ scores for monolingual (xx→xx) summarization on the XL-Sum dataset.
> > >
> > > Beyond Weight Tying: Learning Joint Input-Output Embeddings for Neural Machine Translation (Pappas et al., WMT 2018)
> > >
> > > XL-Sum: Large-Scale Multilingual Abstractive Summarization for 44 Languages (Hasan et al., Findings 2021)
> > >
> > > By Tying Embeddings You Are Assuming the Distributional Hypothesis (Bertolotti & Cazzola, ICML 2024)
> > >
> > > > (ii) The discussion on recent LLMs isn't convincing without accompanying concrete results. In lines 51-60, the paper emphasizes benefits in both performance improvement and inference efficiency. Do these advantages hold for recent, large-vocabulary frontier models?
> > >
> > > Our method's strength lies in coverage. While recent models are strong in languages like Hindi, Russian, and Thai, they still handle under-represented languages such as Coptic and Inuktitut by splitting them into byte tokens. Our approach aims specifically to overcome this fundamental vocabulary limitation.
> > >
> > > We clearly state that our experiments are based on English-centric models (L5-6); we will discuss this limitation further in the revised version of the paper.
> > >
> > > > (iii) Finally, I find the reliance on two papers that did not experiment with decoder-based LMs to back the discussion on the effect of mean initialization unconvincing. This specific point requires support from more relevant papers that use decoder-based LMs.
> > >
> > > We hope that the reviewer will find the empirical results presented in Table R.2 convincing. We find that our conclusions are unchanged when relying on mean-embedding initialisation instead of random.
> > >
> > > |Method|Hi→En|Ru→En|Th→En|*Avg*|
> > > |-|:-:|:-:|:-:|:-:|
> > > | | | **SmolLM-2 360M** | | |
> > > |Random|48.3|53.0|34.8|**45.4**|
> > > |Mean|46.1|52.7|36.5|45.1|
> > > | | |**SmolLM-2 1.7B**| | |
> > > |Random|54.4|56.7|39.4|50.2|
> > > |Mean|55.9|56.1|42.3|**51.5**|
> > >
> > > Table R.2: chrF++ scores comparing random vs. mean embedding initialization. The setup is the same as described in Section 3.

---

> > > > ### Comment · Reviewer_KQmA · 2025-06-09
> > > >
> > > > Thanks for the additional results for (i). It is indeed convincing. I will update the scores accordingly.
> > > >
> > > > Nonetheless, I still think that the paper needs to discuss (ii) in detail showing concrete results, i.e., the extent to which the proposed method alleviates tokenization overfragmentation.

---

### Official Review · Reviewer_QyNG · 2025-04-25

**Rating:** 8
**Confidence:** 5
**Ethics Flag:** 1

**Summary:**

The authors propose a word model that generates an embedding vector from character sequence pixel representations.

There has been some recent work on pixel representations for text (rendered images of text processed as in vision models), this paper uses it as a fallback mechanism that uses a gliding window over 2 characters at a time that get pooled into word vectors that are used instead or alongside regular textual (sub)word embedding vectors. They show superior results compared to byte-based encoding models and vocabulary extensions when fine-tuning an existing language model (SmoLM2-360M/1.7B and Phi-3-Mini) to cover more languages for translating.

**Questions To Authors:**

Question about line 197: "k steps" - how big is a step? one batch? what is the batch size?
Question about Thai: What do you use for word segmentation?

**Reasons To Accept:**

The modeling is quite straightforward. It requires segmenting the text into words, and then encoding each word via pooling of 2-character pixel representation, to be inserted into the embedding layer of the transformer model. It is compared against other standard methods: splitting them up with an existing byte-pair encoding operation set, vocabulary extension, and byte-level encoding. It would have been nice to see two additional baselines: character-level encoding or a subword-word based fallback mechanism and single character pixel model (to have more apples-to-apples comparisons in terms of granularity).

A nice experiment is also the application of this model to code-switched text (Hindi/English) where the Hindi words are encoded with the pixel model and the English words with traditional subword embeddings, showing successful interleaving between the two representation methods.

Overall, this is great work, building on previous ideas of pixel modeling but simpler and used as a fallback mechanism.

**Reasons To Reject:**

None

---

> ### Author Response · Authors · 2025-06-02
> **Reply to Reviewer QyNG**
>
> We thank the reviewer for their time and positive feedback.
> We are especially encouraged by the strong support for our paper to be presented at the conference and are pleased that they found our work to be a valuable extension of previous pixel modeling approaches.
>
> Regarding baselines:
>
> Our decision to focus on a byte-based fallback network as a baseline was motivated by its close alignment with a vocabulary-free approach, which is a core motivation for rendering text as images.
> For this same reason, we did not include a subword-based fallback network, although we agree that the added capacity and a specialized tokenizer would be an interesting direction for future work.
> In contrast, a character-level model would require a much larger vocabulary, such as the 150,000 Unicode codepoints.
> Furthermore, our decision was influenced by the recent literature, which increasingly emphasises byte-level over character-level models.
> Our choice to render 2 characters per image patch directly follows recommendations from previous work by Lotz et al. (2023).
>
> Regarding the questions:
>
> **k steps**: In Table 2, for the cross-lingual transfer experiments, "k steps" refers to the number of update steps, i.e. batches processed. The batch sizes for these experiments are 256 for SmolLM2 models and 512 for Phi-3-mini, as detailed in Table 10 of Appendix A.
>
> **Thai word segmentation**: For Thai, we use DeepCut (Kittinaradorn et al., 2019) for word segmentation (footnote 2, page 4).
>
> Jonas Lotz et al. 2023. Text Rendering Strategies for Pixel Language Models. EMNLP.
>
> Rakpong Kittinaradorn et al. 2019. DeepCut: A Thai word tokenization library using Deep Neural Network.
>
> Please let us know if further clarifications are needed.

---

> > ### Comment · Reviewer_QyNG · 2025-06-02
> >
> > Thank you for the clarification.

---

### Official Review · Reviewer_scWL · 2025-05-13

**Rating:** 7
**Confidence:** 4
**Ethics Flag:** 1

**Summary:**

This paper proposed a vocabulary-free encoder with pixel-level fallback to address the vocabulary constraints in language modeling, by generating input embeddings from text rendered to pixels. The authors conducted substantial multilingual NLP tasks such as machine translation and examined the cross-lingual transfer tasks. The experimental results show that the proposed approach surpasses default tokenization, standard vocabulary expansion and byte-based methods. This approach is applicable to any LLMs and is experimentally shown effective, compared to the larger baseline model.

The paper is well organized and easy to follow. The proposed approach is discussed from various perspectives - model performance in multilingual tasks, text compression, and interleaving text and image representation - and experimentally examined. The results are technically sound, showing robustness across languages and effectiveness of the proposal in each task.


For better readability
- The paper does not seem to follow the COLM paper template. I'd appreciate if you could double-check the webpage (https://colmweb.org/cfp.html), saying "To prepare your submission to COLM 2025, please use the LaTeX style files provided at: https://github.com/COLM-org/Template/archive/refs/tags/2025.zip". Additionally, there is room to improve its readability (Please see the summary for the details.)

**Questions To Authors:**

- Table 1 reports translation performances after one epoch of pretraining and finetuning. Did you try to pick the "best" checkpoint instead and evaluate the performance? I'm curious about whether the proposed approach gets faster converged than the other baselines or so.

**Reasons To Accept:**

Please see the summary.

**Reasons To Reject:**

Overall, this is a good paper with a lot of interesting results of pixel-based fallback.

---

> ### Author Response · Authors · 2025-06-02
> **Reply to Reviewer scWL**
>
> We thank the reviewer for their time and positive feedback.
> We are pleased that they found our work to be technically sound and easy to follow.
>
> We will double-check to ensure the paper follows the COLM template.
>
> We appreciate the suggestion to analyze the "best" checkpoint performance from Table 1 to gain insights into convergence speeds.
> We will include this analysis in the revised version of the paper.
>
> Please let us know if further clarifications are needed.

---

> > ### Comment · Reviewer_scWL · 2025-06-08
> >
> > Thank you for the response. I will keep my score as it is.

---

### Decision · Program_Chairs · 2025-07-08

**Decision:**

Accept

**Comment:**

This paper proposes to augment language models with a fallback network that encodes text based on pixels. The authors show that this approach substantially improves machine translation performance of English-centric language models (e.g., SmolLM2) compared against several baselines.

All reviewers positively highlighted the straightforward yet effective approach, the extensive experiments, and the convincing results. I agree -- this is a great paper that should be published.

A minor drawback of the paper is that it focuses on English-centric models, while many popular language models (e.g., Gemma) are multilingual. As mentioned by reviewer KQmA, it would be desirable to add experiments with a representative model to the paper, but if the authors decide not to do so, they should at least add a more explicit discussion of this limitation to the paper.